# M-ary Cyclic Shift Keying Spread Spectrum Underwater Acoustic Communications Based on Virtual Time-Reversal Mirror

**DOI:** 10.3390/s19163577

**Published:** 2019-08-16

**Authors:** Feng Zhou, Bing Liu, Donghu Nie, Guang Yang, Wenbo Zhang, Dongdong Ma

**Affiliations:** 1Acoustic Science and Technology Laboratory, Harbin Engineering University, Harbin 150001, China; 2Key Laboratory of Marine Information Acquisition and Security (Harbin Engineering University), Ministry of Industry and Information Technology, Harbin 150001, China; 3College of Underwater Acoustic Engineering, Harbin Engineering University, Harbin 150001, China

**Keywords:** spread spectrum communication, time-reversal mirror, underwater acoustic communication

## Abstract

Underwater acoustic communications are challenging because channels are complex, and acoustic waves when propagating in the ocean are subjected to a variety of interferences, such as noise, reflections, scattering and so on. Spread spectrum technique thus has been widely used in underwater acoustic communications for its strong anti-interference ability and good confidentiality. Underwater acoustic channels are typical coherent multipath channels, in which the inter-symbol interference seriously affects the performance of underwater acoustic communications. Time-reversal mirror technique utilizes this physical characteristic of underwater acoustic channels to restrain the inter-symbol interference by reconstructing multipath signals and reduce the influence of channel fading by spatial focusing. This paper presents an M-ary cyclic shift keying spread spectrum underwater acoustic communication scheme based on the virtual time-reversal mirror. Compared to the traditional spread spectrum techniques, this method is more robust, for it uses the M-ary cyclic shift keying spread spectrum to improve the communication rate and uses the virtual time-reversal mirror to ensure a low bit error rate. The performance of this method is verified by simulations and pool experiments.

## 1. Introduction

Underwater acoustic (UWA) communication technique so far is the only medium to communicate over long distances in water. However, it offers more challenges than conventional terrestrial wireless communications [1]. The complicated marine environment causes various kinds of interferences in UWA channels [2,3]. For lower frequencies, noise in the seawater is random, inconsistent and frequency varying while for higher frequencies the acoustic signals suffer high attenuation; thus, there is only a limited band of frequencies for UWA communications. Besides, the acoustic signals reach the receiver along different paths due to the interface, reflections, refractions, random scattering, etc. These factors lead to distortion of the received signals and degradation of the performance of UWA communication systems.

To cope with challenging UWA channels, researchers have proposed several algorithms and communication techniques [4,5,6,7]. For the multipath interference and the low signal-to-noise ratio (SNR) in UWA communications, a commonly used technique is the spread spectrum communication [8,9]. The spread spectrum method is a type of transmission, in which the bandwidth occupied by the signal exceeds the minimum bandwidth required to transmit information [10]. Traditional spread spectrum communications include Direct Sequence Spread Spectrum (DSSS), Frequency Hopping Spread Spectrum (FHSS) and Time Hopping Spread Spectrum (THSS), where DSSS is used more commonly due to its resistance to frequency selective fading [11]. DSSS technique promises high reliability but at the cost of a lower communication rate because it uses one sequence to spread only 1 bit of information [12]. In order to improve the performance of the DSSS system, a new M-ary cyclic shift keying spread spectrum (MCSK) technique is thus proposed, which can not only guarantee the high reliability but can also improve the communication rate to a significant extent.

The MCSK is the combination of two techniques that have been used in UWA communications, namely, M-ary spread spectrum modulation and cyclic shift keying (CSK) modulation [13]. In M-ary modulation, multiple spread spectrum sequences are used to represent different information, and each spread spectrum sequence carries more information [14,15]. CSK technique uses the phases of cyclic shift sequences to transmit information. The amount of information contained in each cyclic shift sequence depends on the length of the sequence. The combination of M-ary and CSK can greatly increase the amount of information carried by one sequence, so the MCSK method can significantly improve the spread spectrum gain on communication rates.

Aiming at the influence of multipath interference of UWA channels on communications, Reference [16] first applies the time-reversal mirror (TRM) technique to UWA communications. In order to meet the requirement of low power consumption, we mainly analyze single-element TRM for unidirectional transmission in this paper. By theoretical derivation and simulations, we analyze the focusing gain and applicable conditions of single-element TRM in detail. Moreover, we propose to combine virtual time-reversal mirror (VTRM) with the MCSK technique in UWA communications. Compared with the system proposed in Reference [17], the bit error rate (BER) performance of our system is much better in UWA multipath channels. This MCSK communication scheme based on VTRM (MCSK-VTRM) can significantly improve both the availability and reliability of information transmission.

The content of this paper is organized as follows: Section 2 introduces the MCSK communication technique and analyses its performance. Section 3 introduces how to use TRM to equalize UWA channels and analyzes the influence of multipath structures of channels on TRM. Besides, we introduce the VTRM technique and how to estimate UWA channels with matching pursuit (MP) and base pursuit de-noising (BPDN) in VTRM. Section 4 analyzes the performance of MCSK-VTRM in UWA communications by simulations and experiments. The last part concludes the research contribution of this paper.

## 2. MCSK Communication Technique

### 2.1. Principle of MCSK

The MCSK communication method combines M-ary modulation with CSK modulation. In the M-ary spread spectrum, the transmitter generates several spread spectrum sequences of the same length. The binary data are grouped first, and the amount of information in each group is $\log_{2}M$ bits, where M is the number of sequences. Different spread spectrum sequences modulate different groups of information. Therefore, the amount of information carried by one sequence increases from 1 bit to $\log_{2}M$ bits. In the CSK method, the transmitter generates only one spread spectrum sequence. CSK can modulate multiple bits of information when the generated spread spectrum sequence is cyclically shifted. It uses the cyclic shift of the spread spectrum sequence to map the information [18]. Namely, it uses the phases of spread spectrum sequences to carry information. Therefore, the length of the spread spectrum sequence affects the amount of information contained in each sequence. 

Pseudo-random sequences have good correlation characteristics, so they are often used as spread spectrum sequences to carry information. There is a series of spread spectrum sequences of the same length as required at both the transmitter and the receiver in the MCSK system. Assuming that M-ary modulation carries $k_{0}$ bits of information, CSK modulation carries $k_{1}$ bits of information. The spread spectrum sequence generator is implemented by linear shift registers, and it generates multiple spread spectrum sequences $c_{i}(t)$, 1≤i≤M, where M is the number of spread spectrum sequences. Then, the spread spectrum sequence selector cyclically shifts $c_{i}(t)$ to obtain $c_{i,j}(t)$, 1≤j≤N, where N is the length of the sequence. Figure 1 shows the MCSK communication scheme. Firstly, the binary data stream is converted from serial to parallel, that is, each $\left(k_{0}+k_{1}\right)$ bits of information is divided into a group. M-ary modulation of the first $k_{0}$ bits of information is used to decide which $c_{i}(t)$ to choose, and CSK modulation of the last $k_{1}$ bits of information is used to determine the number of cyclic shifts of $c_{i}(t)$. Therefore, one cyclic shift sequence $c_{i,j}(t)$ can carry $\left(k_{0}+k_{1}\right)$ bits of information in MCSK modulation. Compared with DSSS modulation, the amount of information carried by one sequence has been greatly increased, and the signal modulated by the carrier is
(1)$$s(t)=Ac_{i,j}(t)\cos(2\pi f_{c}t+\varphi_{0})$$
where A is the amplitude of the transmitted signal, $f_{c}$ is the frequency of the carrier and $\varphi_{0}$ is the phase of the carrier. After passing through UWA channels, s(t) is affected by multipath fading and additive noise n(t). If the propagation delay of the multipath signal is $\tau_{l}$, 1≤l≤L, where L is the number of multipath and the amplitude of the multipath signal at the receiver is $A_{l}$. The received signal is then expressed as
(2)$$s_{r}(t)=\sum_{l=1}^{L}A_{l}c_{i,j}(t+\tau_{l})cos(2\pi f_{c}t+\varphi_{l})+n(t)$$

At the receiver, spread spectrum sequences are also obtained by the sequence selector and cyclic shift, then the received signal is despread by these spread spectrum sequences and demodulated by the local carrier. Therefore, we can get a matrix $V_{k,m}(t)$, which is the MCSK integral judgments within the duration of one spread spectrum sequence.
(3)$$\begin{array}{ll} V_{k,m}(t) & =\overset{T+\tau_{1}}{\underset{\tau_{1}}{\int }}s_{r}(t)c_{k,m}(t-\tau_{1})\cos(2\pi f_{c}t+\varphi_{0})dt\\ & =\frac{1}{2}A_{1}R_{k,m}(0)+\frac{1}{2}\overset{L}{\underset{l=2}{\sum }}A_{l}R_{k,m}(\tau_{l}-\tau_{1})\cos\left[2\pi f_{c}(\tau_{l}-\tau_{1})\right]\\ & +\overset{T+\tau_{1}}{\underset{\tau_{1}}{\int }}n(t)c_{k,m}(t-\tau_{1})\cos(2\pi f_{c}t+\varphi_{0})dt \end{array}$$
where $R_{k,m}(0)$ is
(4)$$R_{k,m}(0)=\overset{T+\tau_{1}}{\underset{\tau_{1}}{\int }}c_{k,m}(t-\tau_{1})c_{i,j}(t-\tau_{1})dt$$
and $R_{k,m}(\tau_{l}-\tau_{0})$ is
(5)$$R_{k,m}(\tau_{l}-\tau_{1})=\overset{T+\tau_{1}}{\underset{\tau_{1}}{\int }}c_{k,m}(t-\tau_{1})c_{i,j}(t-\tau_{l})dt$$

As shown in Figure 2, $V_{k,m}(t)$ varies with k and m, where k represents the sequence number and m represents the cyclic shift number. Because of the good autocorrelation characteristic of spread spectrum sequences, the spread spectrum sequence corresponding to the transmitted information can be found by searching for the maximum value in $V_{k,m}(t)$. After the received signal is despread by the corresponding sequence and demodulated by the local carrier, the original information is restored by parallel to serial conversion.

### 2.2. Performance Analysis

Compared with the traditional spread spectrum, MCSK improves the communication rate by increasing the amount of information modulated by each spread spectrum sequence. In MATLAB, we simulated and compared the BER performances of the DSSS system, CSK spread spectrum system, M-ary spread spectrum system and MCSK system in the additive white Gauss noise (AWGN) channel when the length of spread spectrum sequences in all these systems was 15, and all the spread spectrum sequences were composed of kasami sequences and their cyclic shift sequences. We set $f_{c}$ to 10 kHz, the communication bandwidth to 4 kHz and the sampling rate to 48 kHz. Each system transmitted 1000 bits of data each time and repeated 200 experiments. As known in the DSSS system, one spread spectrum sequence carries only 1 bit of information. In the M-ary spread spectrum system, there were four spread spectrum sequences, so each of them could carry 2 bits of information. In the CSK system, a spread spectrum sequence was cyclically shifted to eight sequences, and each cyclic shift sequence could carry 3 bits of information. In the MCSK system, there were thirty-two spread spectrum sequences in total, and each of them could carry 5 bits of information. When the chip widths of four spread spectrum systems were all 0.5 ms, the communication rates of DSSS, CSK, M-ary system, and MCSK system were as shown in Table 1.

From the BER curves of these four systems in Figure 3, it can be seen that the anti-noise performances of them are arranged from high to low in order: DSSS system, M-ary spread spectrum system, CSK spread spectrum system, MCSK system. Compared with DSSS, the other three methods change from the single sequence decision to the multi-sequence decision, which results in the cross-correlation interference of othefr sequences. In the other three systems, the increase in communication rates is at the cost of reducing reliability. The total number of spread spectrum sequences in the MCSK system is the largest, that is, the cross-correlation interference at the receiver is the most serious, so the anti-noise performance of the MCSK system is the worst.

When we take the communication rates as the reference standard, the performances of these four systems are simulated and compared in the AWGN channel again. The chip widths of four spread spectrum systems were all 0.5 ms. In the M-ary spread spectrum system, one sequence carried 2 bits of information. In the CSK system, one sequence carried 4 bits of information, and in the MCSK system, one sequence carried 8 bits of information. Table 2 shows the communication rates and the length of the spread spectrum sequences of the four systems.

As shown in Figure 4, in the communication BER of $10^{-4}$ magnitude, the anti-noise performance of MCSK system is 1.5 dB higher than that of the CSK system, 2.8 dB higher than that of M-ary spread spectrum system and 4.2 dB higher than that of DSSS system, which is related to the larger spread-spectrum gain of the longer spread spectrum sequence.

## 3. TRM Technique

### 3.1. Theory of TRM

TRM has a good space-time focusing performance and has attracted wide attention in UWA communications [19,20,21,22]. The theoretical basis of TRM is the reciprocity of the acoustic field. It can match UWA channels automatically, thus suppressing inter-symbol interference and channel fading [23]. If the signal is transmitted from a distant acoustic source, then the received signal in the receiver shows a complex structure, which contains the information of the UWA channel. When the received signal is time-reversed and retransmitted to the ocean, all multipath signals reach the acoustic source at the same time, thus realizing the energy aggregation of multipath signals.

The single-element TRM is composed of only one hydrophone array. Compared with the TRM arrays, it simplifies the complexity of the equipment and therefore open wide range of applications in UWA communications. Now the focusing gain of single-element TRM is analyzed concretely. It is assumed that the transmitted detection signal is p(t), and the channel impulse response (CIR) of the UWA channel is h(t). After passing through the UWA channel, the detection signal arriving at the receiver is
(6)$$p_{r}(t)=p(t)⊗h(t)+n(t)=\sum_{i=1}^{L}\left[A_{i}p(t-\tau_{i})+n_{i}(t)\right]$$

Assuming that the noise $n_{i}(t)$ of each path is independent of each other, and the SNRs are the same, the noise of each path satisfies
(7)$$E[n_{i}(t)n_{j}(t)]=0\;(i\neq j)$$

The SNRs of each path is
(8)$$SNR_{in}=\frac{A_{1}^{2}}{\sigma_{1}^{2}}=\frac{A_{2}^{2}}{\sigma_{2}^{2}}=\frac{A_{3}^{2}}{\sigma_{3}^{2}}=\ldots =\frac{A_{L}^{2}}{\sigma_{L}^{2}}.$$
where $\sigma_{i}^{2}$ is the variance of the noise superimposed on the signal in the path i. When the received signal is processed by TRM, the components of each multipath signal are superposed in the same phase at the same time. Then, the reconstructed received signal can be expressed as
(9)$$p_{r}^{\prime }(t)=\sum_{i=1}^{L}A_{i}p(t)+\sum_{i=1}^{L}n_{i}(t)$$

Therefore, the SNR of the time-reversed signal can be deducted as
(10)$$\begin{array}{ll} SNR_{TRM} & =(\sum_{i=1}^{L}A_{i}{)}^{2}/\sum_{i=1}^{L}\sigma_{i}^{2}=(\sum_{i=1}^{L}A_{i}{)}^{2}/(\frac{\sigma_{1}^{2}}{A_{1}^{2}}\sum_{i=1}^{L}A_{i}^{2})\\ & =\frac{A_{1}^{2}}{\sigma_{1}^{2}}(\sum_{i=1}^{L}A_{i}{)}^{2}/\sum_{i=1}^{L}A_{i}^{2}=\frac{A_{1}^{2}}{\sigma_{1}^{2}}(\sum_{i=1}^{L}A_{i}^{2}+\sum_{i=1}^{L}\sum_{\begin{array}{l} j=1\\ j\neq i \end{array}}^{L}A_{i}A_{j})/\sum_{i=1}^{L}A_{i}^{2}\\ & =\frac{A_{1}^{2}}{\sigma_{1}^{2}}[1+(\sum_{i=1}^{L}\sum_{\begin{array}{l} j=1\\ j\neq i \end{array}}^{L}A_{i}A_{j})/\sum_{i=1}^{L}A_{i}^{2}]=SNR_{in}+\frac{A_{1}^{2}}{\sigma_{1}^{2}}(\sum_{i=1}^{L}\sum_{\begin{array}{l} j=1\\ j\neq i \end{array}}^{L}A_{i}A_{j})/\sum_{i=1}^{L}A_{i}^{2} \end{array}$$

In decibels, it can be written as
(11)$$SNR_{TRM}=SNR_{in}+10\mathrm{lg}[1+(\sum_{i=1}^{L}\sum_{\begin{array}{l} j=1\\ j\neq i \end{array}}^{L}A_{i}A_{j})/\sum_{i=1}^{L}A_{i}^{2}]$$

It can be seen that after TRM treatment, the multipath diversity in the received signal is realized, that is, the SNR of the time-reversed signal is larger than that of the original received signal. $10\mathrm{lg}[1+(\sum_{i=1}^{L}\sum_{\begin{array}{l} j=1\\ j\neq i \end{array}}^{L}A_{i}A_{j})/\sum_{i=1}^{L}A_{i}^{2}]$ in (11) is the TRM focusing gain, which is closely related to the number and amplitude of multipath signals. In UWA communications, the sparser the channel, the greater the focusing gain obtained by using single-element TRM.

### 3.2. Influence of Multipath Structure on TRM

The focusing effect of TRM is related to the multipath structure of UWA channels. Taking the single-element TRM as an example, and the CIR of the UWA channel is assumed to be

(12)$$h(t)=A_{1}\delta (t)+\sum_{m=2}^{L}A_{m}\cdot \delta (t-\tau_{m})$$

After TRM processing, the matched output of CIR is

(13)$$\begin{array}{ll} h(t)⊗h(-t) & =\sum_{m=1}^{L}A_{m}^{2}\cdot \delta (t)+\sum_{m=2}^{L}A_{1}A_{m}\left[\delta (t-\tau_{m})+\delta (t+\tau_{m})\right]\\ & +\sum_{m=2}^{L}\sum_{n=m+1}^{L}A_{m}A_{n}\left[\delta (t-(\tau_{m}-\tau_{n}))+\delta (t+(\tau_{m}-\tau_{n}))\right]. \end{array}$$

Therefore, after TRM treatment, the maximum amplitude is increased to
(14)$$A_{\max}=\sum_{m=1}^{L}A_{m}^{2}.$$

It can be seen from (14) that the number and magnitude of multipath in the UWA channel will affect the magnitude of the maximum of the focusing peak. Next, the sidelobe structure of the time-reversal channel is analyzed. The unilateral sidelobe interference is
(15)$$B+C=\sum_{m=2}^{L}A_{1}A_{m}\delta (t-\tau_{m})+\sum_{m=2}^{L}\sum_{n=m+1}^{L}A_{m}A_{n}\delta (t+(\tau_{m}-\tau_{n})).$$

Next, taking six paths of a multipath channel shown in Figure 5 as an example, the effect of multipath delay on the performance of single-element TRM is analyzed.

When the number of multipath L is 6,
(16)$$\begin{array}{ll} B & =\sum_{m=2}^{6}A_{1}A_{m}\delta (t-\tau_{m})\\ & =A_{1}A_{2}\delta (t-\tau_{2})+A_{1}A_{3}\delta (t-\tau_{3})+A_{1}A_{4}\delta (t-\tau_{4})+A_{1}A_{5}\delta (t-\tau_{5})+A_{1}A_{6}\delta (t-\tau_{6})\\ & =A_{1}A_{2}\delta (t-\Delta t_{1})+A_{1}A_{3}\delta (t-(\Delta t_{1}+\Delta t_{2}))+A_{1}A_{4}\delta (t-(\Delta t_{1}+\Delta t_{2}+\Delta t_{3}))\\ & +A_{1}A_{5}\delta (t-(\Delta t_{1}+\Delta t_{2}+\Delta t_{3}+\Delta t_{4}))+A_{1}A_{6}\delta (t-(\Delta t_{1}+\Delta t_{2}+\Delta t_{3}+\Delta t_{4}+\Delta t_{5})), \end{array}$$
and
(17)$$\begin{array}{ll} C & =\sum_{m=2}^{6}\sum_{n=m+1}^{6}A_{m}A_{n}\delta (t+(\tau_{m}-\tau_{n}))\\ & =A_{2}A_{3}\delta (t-\Delta t_{2})+A_{2}A_{4}\delta (t-(\Delta t_{2}+\Delta t_{3}))+A_{2}A_{5}\delta (t-(\Delta t_{2}+\Delta t_{3}+\Delta t_{4}))\\ & +A_{2}A_{6}\delta (t-(\Delta t_{2}+\Delta t_{3}+\Delta t_{4}+\Delta t_{5}))+A_{3}A_{4}\delta (t-\Delta t_{3})+A_{3}A_{5}\delta (t-(\Delta t_{3}+\Delta t_{4}))\\ & +A_{3}A_{6}\delta (t-(\Delta t_{3}+\Delta t_{4}+\Delta t_{5}))+A_{4}A_{5}\delta (t-\Delta t_{4})+A_{4}A_{6}\delta (t-(\Delta t_{4}+\Delta t_{5}))\\ & +A_{5}A_{6}\delta (t-\Delta t_{5}). \end{array}$$

If the structure of the multipath channel is assumed as shown in Figure 6a, where $\Delta t_{1}\neq \Delta t_{2}\neq \Delta t_{3}\neq \Delta t_{4}\neq \Delta t_{5}$, the delays of each item in C are different and independent, and they cannot be superimposed on each item in B. Therefore, in such multipath channel, only the energy of the direct path becomes larger after TRM treatment, and the primary-secondary peak ratio of the channel is significantly improved.

If the structure of the multipath channel is assumed as shown in Figure 7a, where $\Delta t_{1}=\Delta t_{2}=\Delta t_{3}=\Delta t_{4}=\Delta t_{5}$, each item in C is added to each item in B, which will increase the energy of all paths. The primary–secondary peak ratio of this matched channel is equal to the original channel. Therefore, TRM is not suitable for similar structure channels.

### 3.3. Virtual Time-Reversal Mirror

In the VTRM technique, the signals are one-way transmission in the UWA channel [24]. VTRM uses the received detection signal to estimate the channel and reduces the complexity of communication systems. The diagram of the VTRM is shown in Figure 8, and the algorithm of VTRM is shown in Algorithm 1. After passing through the UWA channel, the received detection signal is used to get the CIR $h^{\prime }(t)$ with the appropriate channel estimation method at the receiver. Then, the convolution of the received signal and the estimated time-reversal channel is completed.

The transmitted signal which passes through the UWA channel can be expressed as
(18)$$s_{r}(t)=s(t)⊗h(t)+n(t).$$

After VTRM treatment, the signal obtained is as follows
(19)$$\begin{array}{ll} r(t) & =s_{r}(t)⊗h^{\prime }(-t)\\ & =[s(t)⊗h(t)]⊗h^{\prime }(-t)+n(t)⊗h^{\prime }(-t)\\ & =s(t)⊗[h(t)⊗h^{\prime }(-t)]+n(t)⊗h^{\prime }(-t). \end{array}$$
**Algorithm 1:** Algorithm of VTRM.1: **Input:** The received signal2: Intercept the detection sequence and the information sequence from the received signal which passes through the UWA channel.3: Use the detection sequence to estimate the UWA channel with an appropriate method.4: Make time-reversal of the estimated channel.5: Make the convolution of the information sequence with the estimated time-reversal channel.6: **Output:** The information signal after VTRM

In (19), the matched channel $h(t)⊗h^{\prime }(-t)$ can be regarded as an effective channel, through which the signal is transmitted and is basically the correlation between the CIR of the actual UWA channel and the CIR of the estimated UWA channel. When the UWA channel is estimated accurately and the time delay of each multipath is independent, the energy of the direct path is much higher than that of other paths in the matched channel. VTRM realizes the energy in-phase superposition of the multipath signals, which can suppress the inter-symbol interference caused by multipath effect and enhance the output SNR.

Equation (19) can also be regarded as the convolution of received multipath signals s(t)⊗h(t) and the time-reversal channel $h^{\prime }(-t)$, which is equivalent to the superposition of received multipath signals when the direct path is taken as reference, to achieve energy aggregation. Besides, the white noise n(t) is not correlated with $h^{\prime }(-t)$, so the energy of n(t) is not improved even after TRM treatment. Therefore, the output SNR is improved.

### 3.4. Methods of Channel Estimation in VTRM

#### 3.4.1. MP Algorithm

The focusing effect of VTRM is related to the accuracy of UWA channel estimation. When the channel estimation is not precise, the poor focusing effect will deteriorate the performance of the system when using the VTRM technique. Consequently, improving the accuracy of channel estimation is the key to improve the performance of the VTRM communication system. When a signal is sparse or compressible, we can obtain the condensed representation of the compressed signal by a linear projection, and the data can reconstruct the original digital signal in an undistorted or low distortion way. It is known to us that CIR of UWA channels are sparse, that is, the energy of UWA channels is mainly concentrated in a few paths while most of the paths are zero or very small. Due to this characteristic, UWA channels can be modeled by compressed sensing theory.

A detection signal should be first transmitted to pass through the UWA channel to estimate the CIR, and the received detection signal is expressed as
(20)y(n)=x(n)⊗h(n)+v(n).

Suppose the length of detected signals is N, and the length of the UWA channel is L. Then the above formula can be expressed as
(21)$$\left[\begin{array}{c} y(0)\\ y(1)\\ \vdots \\ y(N-2)\\ y(N+L-1) \end{array}\right]=\left[\begin{array}{c} x(0)\\ x(1)\\ \vdots \\ 0\\ 0 \end{array}\right]h(0)+\left[\begin{array}{c} 0\\ x(0)\\ \vdots \\ 0\\ 0 \end{array}\right]h(1)+\cdots +\left[\begin{array}{c} 0\\ 0\\ \vdots \\ x(N-1)\\ x(N) \end{array}\right]h(L-1)+\left[\begin{array}{c} v(0)\\ v(1)\\ \vdots \\ v(N+L-2)\\ v(N+L-1) \end{array}\right].$$

It can also be written as
(22)$$y(n)=\sum_{l=0}^{L-1}x_{l}h(l)+v(n).$$

It may be noted that the received signal is regarded as a linear combination of the detection signal [25]. If sparse decomposition of the received signal is carried out, the CIR can be obtained by the compressive sensing reconstruction algorithm.

The MP algorithm is one of the most basic methods for sparse signal reconstruction. In this algorithm, the sparse coefficient of the CIR of the UWA channel is obtained by optimizing the objective function, and the CIR is decomposed into the combination of some atoms in the dictionary. For specific steps of implementation of this algorithm please refer to Reference [26]. First, the received signal is selected as residual. In each iteration process, the atom which matches the residual best is selected from the dictionary to estimate the component of CIR, meanwhile the residual is update. When the residual satisfies the threshold, the iteration process stops.

#### 3.4.2. BPDN Algorithm

BPDN algorithm is an effective algorithm to solve the convex optimization problem by linear programming. It is the improvement of the basis pursuit algorithm to adapt to the noise.

The sparsity of the UWA channel can be described by the number of non-zero elements in CIR. This problem can be solved by minimizing $l_{0}$ norms
(23)$$\min{\left\|x\right\|}_{0},\;s.t.\;y=Ax+v.$$

But (23) is a problem of non-convex function optimization. In Reference [27], Chen, Donoho, and Saunders pointed out that the minimization *l*_1_ optimization problem is equivalent to the minimization $l_{0}$ optimization problem under certain conditions, so the problem is changed into
(24)$$\min{\left\|x\right\|}_{1},\;s.t.\;y=Ax+v.$$

In the BPDN algorithm, the problem of sparse signal with noise can be optimized as follows
(25)$$\underset{x}{\min}\frac{1}{2}{\left\|y-Ax\right\|}_{2}^{2}+\lambda {\left\|x\right\|}_{1},$$
where λ is the regularization coefficient and controls the balance between permissible error and sparsity. It is important to select appropriate λ to achieve accurate signal reconstruction. According to Reference [28], spaRSA is suitable for solving these $l_{2}-l_{1}$ problems, which is quicker to reconstruct sparse UWA channels than the original BPDN algorithm.

## 4. Analysis of the MCSK–VTRM System

### 4.1. Simulations

This part studies the improvement of reliability after VTRM is applied to the MCSK system. The CIR of the UWA channel used in the simulation is shown in Figure 9. It can be seen that VTRM is suitable for this multipath channel to achieve channel equalization. The peak-sidelobe ratio of the matched channel is better than that of the original channel. That is, the energy of the direct path is enhanced while the energy of other paths is relatively suppressed, which shows a great channel focusing characteristic.

The simulation conditions are given in Table 3. Assuming that the CIR is known, the reliability of the traditional DSSS system, MCSK system, and MCSK-VTRM system is compared in Figure 10. In all these systems, the spread spectrum sequences are composed of kasami sequences and their cyclic shift sequences. Each system transmits 1000 bits of data each time and repeats 200 experiments. In the MCSK system and the MCSK-VTRM system, one spread spectrum sequence carries 5 bits of information.

In Figure 10, it can be seen that the reliability of the MCSK system is lower than that of the DSSS system. When the BER is $10^{-2}$, the SNR of MCSK is 7 dB higher than that of DSSS. It is known that the increase in the amount of information carried by each spread spectrum sequence improves the effectiveness of the MCSK communication system, but the cross-correlation between different spread spectrum sequences interfere with the decision of the receiver. However, when the VTRM technique is applied to the MCSK system, the reliability of communication is further improved, for VTRM can effectively suppress the multipath interference caused by UWA channels. Therefore, the MCSK-VTRM communication system proposed in this paper can not only improve the transmission rate of traditional spread spectrum communication but also improve the anti-noise ability and anti-multipath interference ability of UWA communication systems. When the length of spread spectrum sequences is kept the same, compared with the hybrid spread spectrum method in the Reference [18], this MCSK-VTRM system only uses a single carrier and can ensure higher efficiency and reliability while the system complexity is lower.

In practice, the CIR of the UWA channel is unknown. To utilize VTRM to compensate the influence of the UWA channel, the received detection signal is first used to estimate the CIR of the UWA channel at the receiver. As shown in Figure 11, a transmitted signal is composed of a synchronous signal, a detection signal, some protection intervals and a spread spectrum signal which carries information. The synchronous signal is usually the linear frequency modulation signal, and the detection signal can choose the linear frequency modulation signal or other training sequences.

The BER curves of the MCSK-VTRM system with the MP algorithm and SpaRSA algorithm based on BPDN estimating CIR are shown in Figure 12. It can be seen that when the SNRs are the same, the BER of the system using BPDN is slightly lower. Compared with BER of MP, that of BPDN is lower about 0.5 dB when BER is $10^{-2}$. The performance of BPDN channel estimation is better than that of MP channel estimation, but the BPDN algorithm has a large amount of computation and a long operation time.

### 4.2. Experiments

To verify the performance of the proposed MCSK-VTRM system, an experiment was also carried out in January 2018 in the channel pool of Harbin Engineering University. The pool is 45 meters long, 6 meters wide and 5 meters deep. It is surrounded by silent wedges and has a sandy bottom. Both the transducer and the receiving hydrophone used in the experiment are omnidirectional. They were placed 2 meters below the water surface, and the horizontal distance was 8.35 meters. We used Cool Edit Pro to send and receive signals, and the structure of the transmitted signal is as shown in Figure 11. Figure 13 shows pictures of experiment equipment and Figure 14 shows the layout of the experiment site.

We used the MP algorithm to estimate the channel used in the experiment. The CIR of the pool channel is shown in Figure 15.

The parameters of pool experiments were kept the same as those in the simulations, which are shown in Table 3. The spread spectrum sequences used in experiments are composed of kasami sequences and their cyclic shift sequences. The frequency band of the transducer was 8-16 kHz, and the transmitted image was a binary image with a size of 27.4 kilobits. In the MCSK system and the MCSK-VTRM system, one spread spectrum sequence carries 5 bits of information. Figure 16 compares the images received by DSSS, MCSK, and MCSK-VTRM when the transmitting power was the same. MCSK-VTRM system used the MP algorithm to estimate the channel. The received image in the DSSS system is the clearest compared to the original image, while the received image in the MCSK system is the blurriest.

Table 4 shows the performance comparison of the three kinds of communications. The BER is calculated by transmitting a large amount of data in the pool in different communication systems. When the length of spread spectrum sequences is the same, the communication rate of MCSK is higher than that of DSSS, but the BER is worse. When VTRM is applied to MCSK, the BER decreases. It is verified that VTRM can effectively equalize UWA channels and suppress multi-path interference, thus improving the accuracy of correct judgment in communications.

In the simulations, the BER of the MCSK-VTRM system is almost similar to that of the DSSS system. However, the performance of the MCSK-VTRM system is not close to that of the DSSS system in the poor experiments. This is because the sparsity of the channel in the experiment is not good enough. As can be seen in Figure 15, the channel in the experiment is multi-path concentrated and mostly concentrated near the main path. But as shown in Figure 9, the sparsity of channel used in the simulation is good, where most channel coefficients have smaller energy. The distribution of several taps with larger energy is far apart, and the time delay difference between each tap is unequal. From the discussion in Section 3, we know that the effect of VTRM is related to the multipath structure of the UWA channel. The multipath structure of the channel in the experiments is not ideal enough and leads to the results that do not reach the ideal state of simulations.

## 5. Conclusions

This paper presents an MCSK-VTRM technique, which is applied to UWA communications. In this communication method, the M-ary technique and the CSK technique are combined to improve the amount of information carried by one spread spectrum sequence, so the efficiency of communications is improved. VTRM is used to equalize UWA channels to improve the reliability of communications and avoid the inter-symbol interference caused by UWA multipath channels. First, this paper establishes an MCSK system model and compares it with traditional spread spectrum communications. Then, we introduce the application of TRM in UWA communications and analyze the focusing performance and applicable conditions of TRM in detail by theoretical formula derivation and MATLAB simulations. Finally, simulations and poor experiments verify that MCSK-VTRM communications can maintain robust reliability and improve the communication rate.

## Figures and Tables

**Figure 1 sensors-19-03577-f001:**
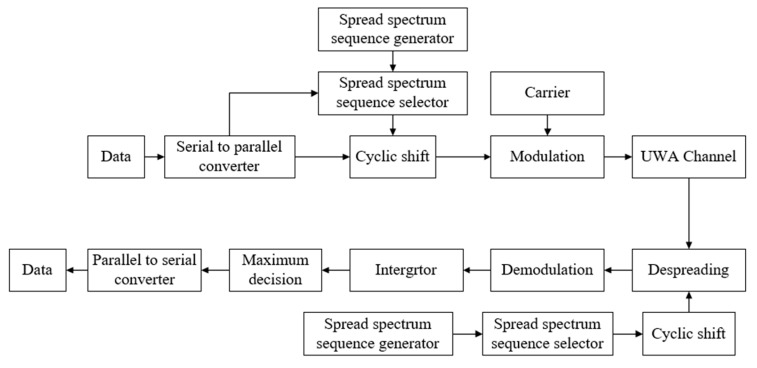
The MCSK communication scheme.

**Figure 2 sensors-19-03577-f002:**
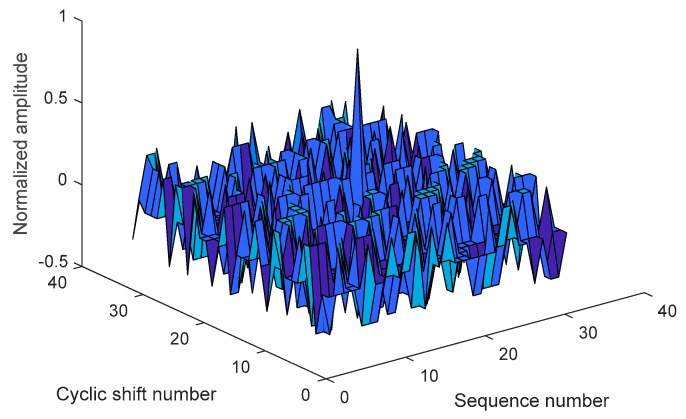
Diagram of MCSK integral judgments.

**Figure 3 sensors-19-03577-f003:**
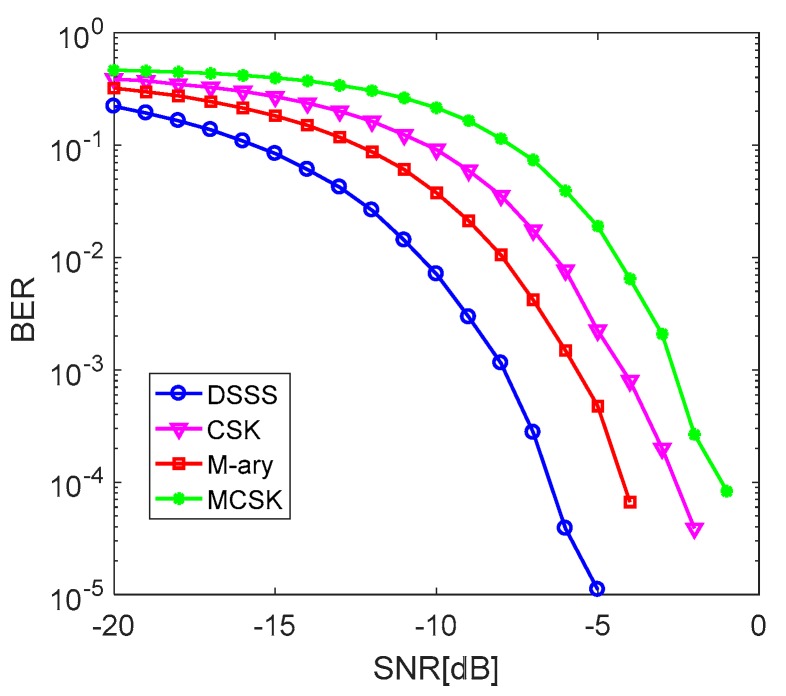
BER curves of four systems with the same length of spread spectrum sequences.

**Figure 4 sensors-19-03577-f004:**
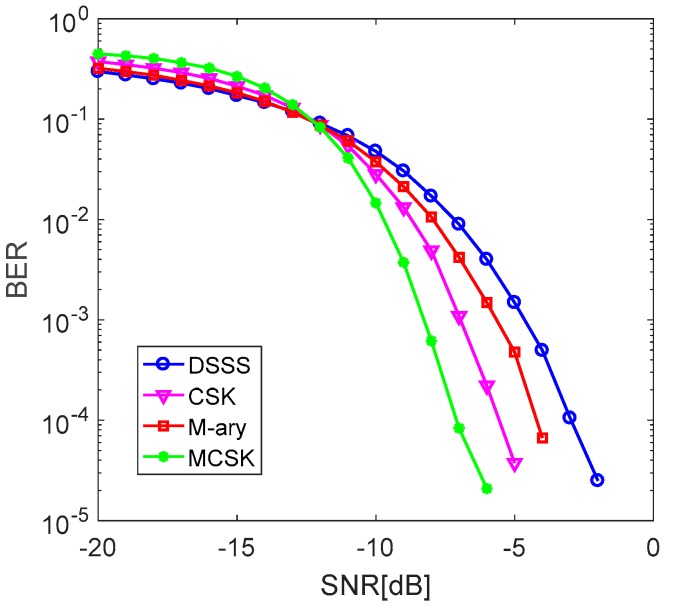
BER curves of four systems with approximately equal communication rates.

**Figure 5 sensors-19-03577-f005:**
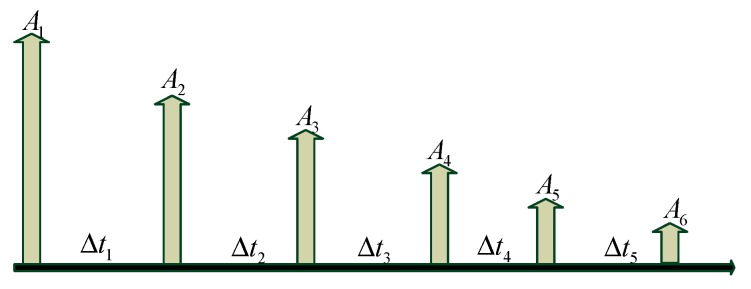
Diagram of the multipath channel.

**Figure 6 sensors-19-03577-f006:**
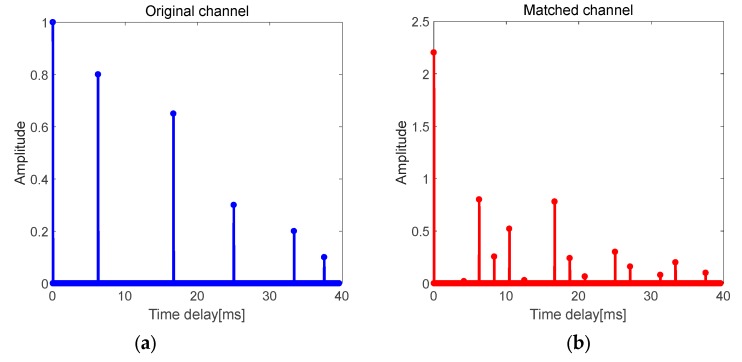
Multipath channel with unequal time delay differences. (**a**) Original channel; (**b**) matched channel.

**Figure 7 sensors-19-03577-f007:**
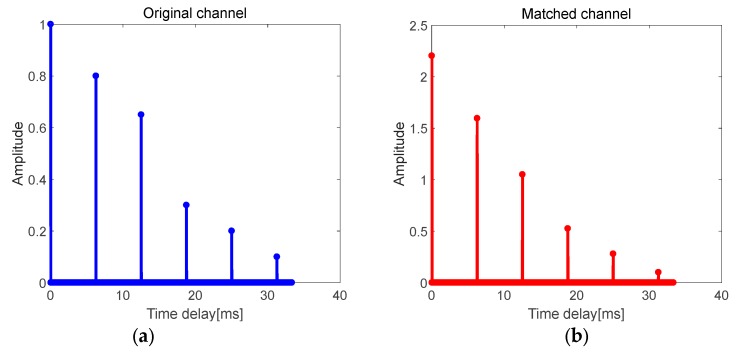
Multipath channel with equal time delay differences. (**a**) Original channel; (**b**) matched channel.

**Figure 8 sensors-19-03577-f008:**
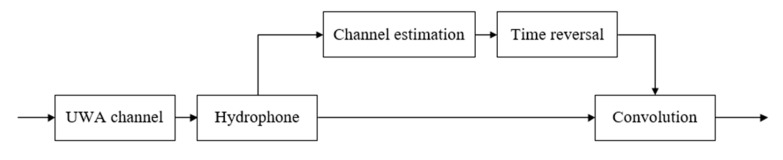
Diagram of VTRM.

**Figure 9 sensors-19-03577-f009:**
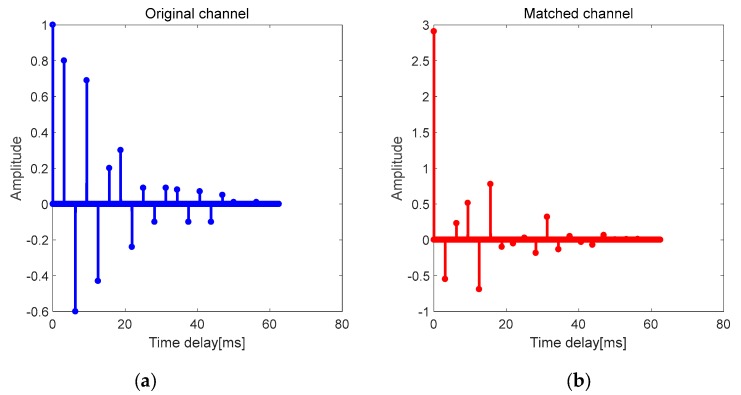
CIR of the UWA channel in simulation. (**a**) Original channel; (**b**) matched channel.

**Figure 10 sensors-19-03577-f010:**
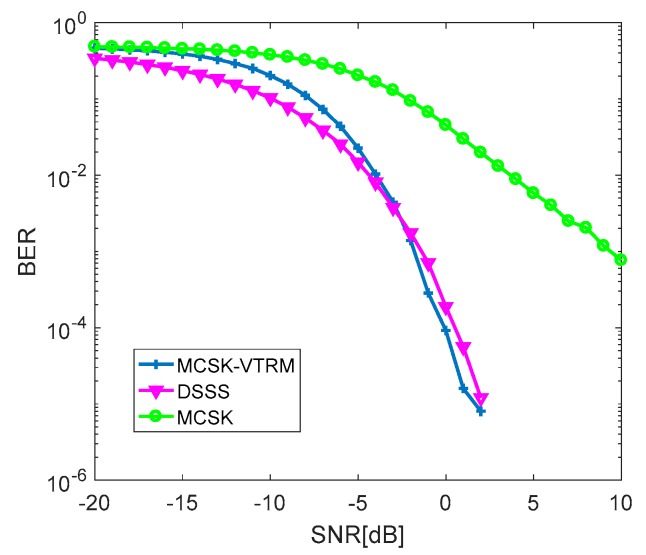
BER Curves of three systems.

**Figure 11 sensors-19-03577-f011:**
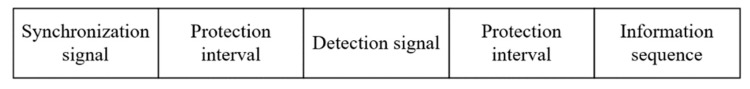
Structure of the transmitted signal.

**Figure 12 sensors-19-03577-f012:**
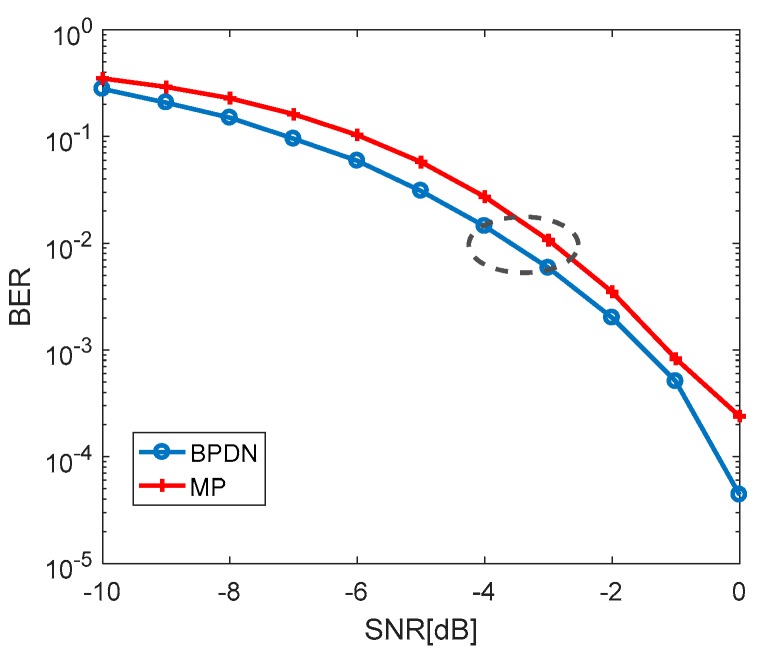
Performance comparison between MP and BPDN.

**Figure 13 sensors-19-03577-f013:**
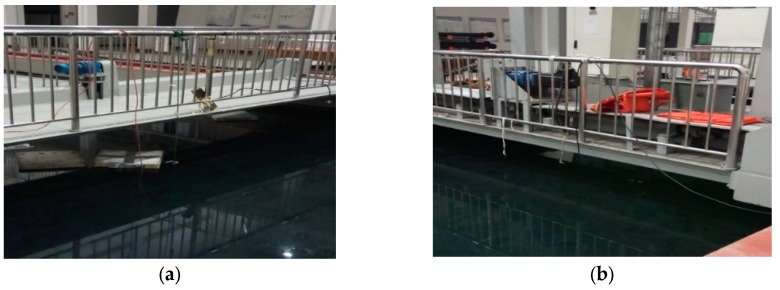
Pictures of experiment equipment. (**a**) The transducer; (**b**) the receiving hydrophone.

**Figure 14 sensors-19-03577-f014:**
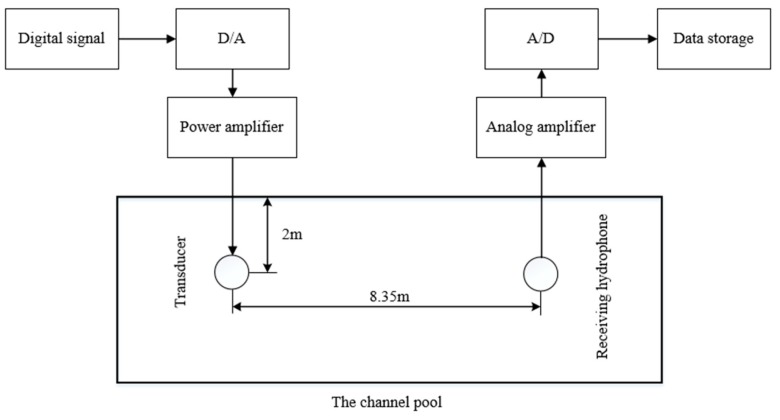
The layout of the experiment site.

**Figure 15 sensors-19-03577-f015:**
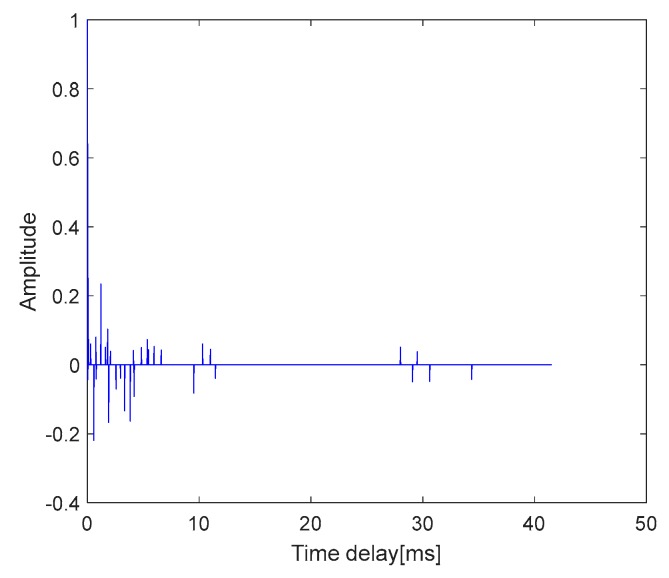
CIR of the pool channel.

**Figure 16 sensors-19-03577-f016:**
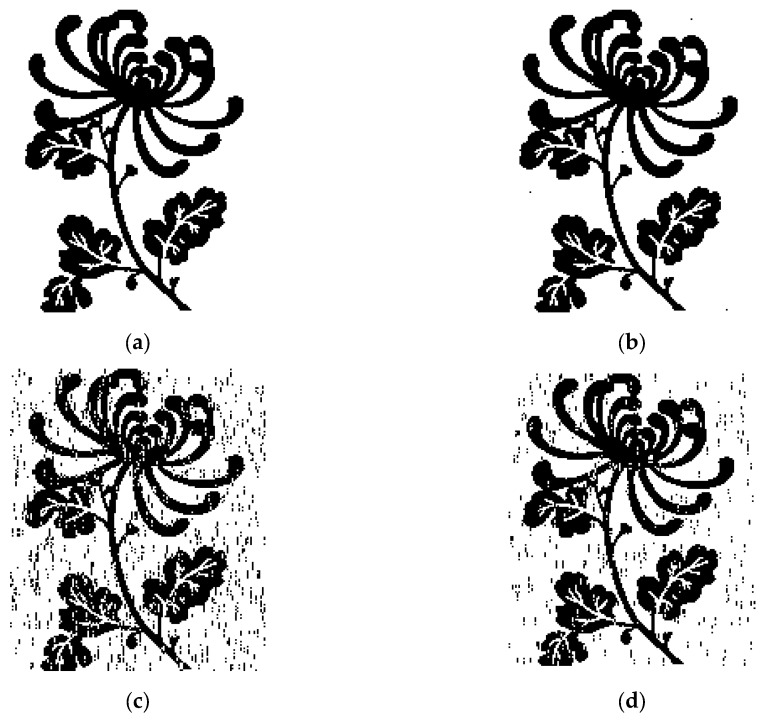
Transmitted and received images. (**a**) Original image; (**b**) received image in DSSS system; (**c**) received image in MCSK system; (**d**) received image in MCSK-VTRM system.

**Table 1 sensors-19-03577-t001:** Communication rates of systems.

Communication System	Communication Rate
DSSS	133.3 bps
M-ary	266.7 bps
CSK	399.0 bps
MCSK	665.0 bps

**Table 2 sensors-19-03577-t002:** Parameters of four systems.

Communication System	Communication Rate	The Length of Spread Spectrum Sequences
DSSS	285.7 bps	7
M-ary	266.7 bps	15
CSK	258.1 bps	31
MCSK	253.9 bps	63

**Table 3 sensors-19-03577-t003:** Simulation parameters.

Parameter Name	Value and Unit
sampling rate	48 kHz
carrier frequency	10 kHz
communication bandwidth	4 kHz
the length of spread spectrum sequences	15

**Table 4 sensors-19-03577-t004:** Performance Comparison.

Communication System	Communication Rate	BER
DSSS	133.3 bps	$1.11\times 10^{-4}$
MCSK	665 bps	$6.63\times 10^{-2}$
MCSK-VTRM	665 bps	$3.04\times 10^{-2}$

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
