# Peer review of "M-ary Cyclic Shift Keying Spread Spectrum Underwater Acoustic Communications Based on Virtual Time-Reversal Mirror"

_sensors, 2019, doi:10.3390/s19163577_

Round 1
Reviewer 1 Report
A M-ary Cyclic Shift Keying spread spectrum is proposed in order to get a balance between the bandwidth efficiency and the robustness, which is very important and difficult in UWA communication environment. This manuscript gives us a feasible solution to this problem. The theoretical analysis of TR gives us a detailed explanation of the SNR gain of TR processing. The performance is testified by pool experiment.
This paper is generally well written and easy to follow. The theory/mathematical modeling, and algorithm description were well presented with correct and clear mathematical derivation.
Comments:
1 In Tab.1., the bit rates of different modulation methods are compared. Please briefly introduce the difference between the M-ary method and the CSK method.
2 Bigger marks should be used in Fig.3.
3 The following minor typos were spotted:
Page1, Abstract, Line15, for channel -> when channel
Page2, Line 48, (THSS), with -> where
Page4, Line 114, the row and column in -> the serial number of rows
Author Response
Thank you for your comments concerning our manuscript entitled “M-ary Cyclic Shift Keying Spread Spectrum Underwater Acoustic Communications Based on Virtual Time-Reversal Mirror” (Manuscript ID: sensors-553158). We quite appreciate your insightful comments. Your comments are of great help to our research and the quality of our paper. We wish to express our heartfelt thanks and gratitude to you again.
We went through all the comments carefully and have made corrections exactly according to your kind comments. Please see the attachment. The main corrections in the manuscript and the response to your comments are as follows:
Point 1: In Tab.1., the bit rates of different modulation methods are compared. Please briefly introduce the difference between the M-ary method and the CSK method.

Response 1: Thank you for your careful work. In our original manuscript, the introduction of the M-ary technique and the CSK technique was not exhaustive enough. We have supplemented the introduction of these two technologies in Section 2.1 and the simulation parameters in Section 2.2. Because of your valuable advice, the revised Section 2 is more complete and easy to follow.
Point 2: Bigger marks should be used in Fig.3.
Response 2: Thank you for your suggestion. We checked all the figures in the original manuscript, and we found some of them have the inappropriate sizes of words and marks. We have adjusted these pictures to make them look clearer.
Point 3: The following minor typos were spotted:
Page1, Abstract, Line15, for channel -> when channel
Page2, Line 48, (THSS), with -> where
Page4, Line 114, the row and column in -> the serial number of rows
Response 3: Thank you for your careful reading. I'm sorry our statement is not clear. In the sentence in Line 15, we wanted to use a conjunction to express causation. So we have changed ‘for’ to ‘because’. And we have corrected the error in Page 2. In Page 4, the expression of the MCSK integral decision is not clear. We have corrected it in Line 133-139. We took your suggestion and examined the whole manuscript carefully. We have also corrected other errors in expression.

Reviewer 2 Report
This paper proposed a communication system that combines M-ary spread spectrum modulation and cyclic shift keying modulation. The information bits are transmitted not only M-ary modulation carriers, but also by CSM modulation carriers. Then, the virtual time-reversal mirror technology is used to suppress inter-symbol interference caused by multipath effect, and Matching Pursuit and Base Pursuit De-Noising algorithms are used for channel estimation. In order to improve of the quality of the paper, the authors should reconsider and clarify the following points:
1. The proposed spread spectrum method should be given in more details, in particular the acquisition step of the spread sequence.
2. The references should be added to the comparisons method so that the audience of the journal could understand it clearly.
3. The comparison results in Fig. 3 and Fig. 15 are not fair because the different communication rate between the proposed method and the comparison methods, and no meaningful conclusion can be obtained by this way.
4. The comparison with DSSS in Fig. 10 is not fair because the proposed method uses channel equalization, while DSSS does not .
5 the references are very old, many recent state of art about underwater communication should be added and reviewed.
e.g.
Digital underwater communication with chaos, Communications in Nonlinear Science and Numerical Simulation, 2019, 73: 14-24
Experimental Wireless Communication Using Chaotic Baseband Waveform, IEEE Transactions on Vehicular Technology, 2019, 68(1): 578-591
Modulation recognition of non-cooperation underwater acoustic communication signals using principal component analysis, Applied Acoustics, 2018, 138: 209-215
Time reversal MFSK acoustic communication in underwater channel with large multipath spread, Ocean Engineering, 2018, 152: 203-209
...
Moreover, the writing of the draft should be checked carefully, such as SBR in Fig. 10 should be SNR? Finally, the presented results need to be elaborated clearly to highlight the novelty of the presented work. In the reviewer’s opinion, the current version of the paper is not enough quality to be published. It needs to be revised before the further consideration.
Author Response
Thank you for your comments concerning our manuscript entitled “M-ary Cyclic Shift Keying Spread Spectrum Underwater Acoustic Communications Based on Virtual Time-Reversal Mirror” (Manuscript ID: sensors-553158). We quite appreciate your insightful comments. Your comments are of great help to our research and the quality of our paper. We wish to express our heartfelt thanks and gratitude to you again.
We went through all the comments carefully and have made corrections exactly according to your kind comments. Please see the attatchment. The main corrections in the manuscript and the response to your comments are as follows:
Point 1: The proposed spread spectrum method should be given in more details, in particular the acquisition step of the spread sequence.
Response 1: Thank you for your careful reading. We are very sorry that the method in Section 2 is not given in detail enough. We have reorganized the content of Section 2. And we supplemented the introduction and simulation parameters. We have examined this Section, which is now more complete and easy to follow.
Point 2: The references should be added to the comparisons method so that the audience of the journal could understand it clearly.
Response 2: Thank you very much. Your suggestion is very important to us. In the previous writings, we did not pay enough attention to the comparison of the proposed method and references. In Section 4, we compared the proposed MCSK-VTRM with the traditional DSSS technique and MCSK technique. We can draw the conclusion that the proposed MCSK-VTRM can not only guarantee a high communication rate, but also maintain the robustness of the system in UWA multipath channels. And we have added the comparison between MCSK-VTRM and the hybrid spread spectrum system proposed in reference “Burst mode hybrid spread spectrum technology for covert acoustic communication” under Figure 10. When the length of spread spectrum sequences is the same, compared with the hybrid spread spectrum method, this MCSK-VTRM communication only uses single carrier. But it can ensure higher efficiency and reliability while the system's complexity is lower. Your valuable suggestion make us realize the importance of comparison between different methods. In future research, we will pay more attention to the comparison between the proposed method and other methods, and analyse their advantages and disadvantages.
Point 3: The comparison results in Fig. 3 and Fig. 15 are not fair because the different communication rate between the proposed method and the comparison methods, and no meaningful conclusion can be obtained by this way. 

Response 3: Thank you for your careful work. The reason why we adopt this comparative approach are as follows: In Figure 3 and Figure 15 of original manuscript, the length of spread spectrum sequences of these systems are kept same, that is, the spread spectrum gains of each systems are guaranteed to be equal. The purpose of this comparison is to show that when the spread spectrum gains are the same, the spread spectrum sequences with the same length carry different amount of information in different systems, which leads to different communication rates. Moreover, due to the different cross-correlation interference of spread spectrum sequences, the reliability of each system is also different. The comparison results at the same communication rates of these systems are given in Figure 4.
Point 4: The comparison with DSSS in Fig. 10 is not fair because the proposed method uses channel equalization, while DSSS does not.
Response 4: Thank you for your comment. In Figure 10, the length of the spread spectrum sequences in different systems are equal for the same reason as in Figure 3. Compared with the DSSS system, the communication rate of the MCSK system is improved, but its reliability is decreased. However, the application of VTRM technique in the MCSK system can improve the reliability under multipath channel conditions. This comparison shows that the combination of MCSK technique and VTRM technique can not only ensure the effectiveness of the spread spectrum system, but also but also maintain the robustness of the system in UWA multipath channels.
Point 5: the references are very old, many recent state of art about underwater communication should be added and reviewed.
Response 5: Thank you for this suggestion and the references provided. The references in our original manuscripts are really too old and insufficient, so we went to look up some references in recent years and added them to our manuscript. Your suggestion is very pertinent. We will pay more attention to the use of references in future writing.
After going through your comments, we examined the whole manuscript carefully and corrected the errors and irregular statements. We really sincerely appreciate and accept all your suggestions, which have a profound impact on our future research and papers. In the future, we will pay attention to the comparison of methods in the experiment, and form good writing habits and styles.

Reviewer 3 Report
There is little detail on the configuration used for the experimental measurements. This probably leads to the confusion I detail below.
The simulation in Figure 10 shows that when the DSSS BER is near 10^-4, the MCSK-VTRM is close to the DSSS performance. However, Figure 15 and Table 5 show that the experiment reveals that theMCSK-VTRM performance is closer to the MCSK performance than the DSSS results.
I think that there needs to be some discussion of the discrepancy between simulation and measurement.
Author Response
Thank you for your comments concerning our manuscript entitled “M-ary Cyclic Shift Keying Spread Spectrum Underwater Acoustic Communications Based on Virtual Time-Reversal Mirror” (Manuscript ID: sensors-553158). We quite appreciate your insightful comments. Your comments are of great help to our research and the quality of our paper. We wish to express our heartfelt thanks and gratitude to you again.
We went through all the comments carefully and have made corrections exactly according to your kind comments. The main corrections in the manuscript and the response to your comments are as follows:
Point 1: There is little detail on the configuration used for the experimental measurements. This probably leads to the confusion I detail below. 

Response 1: Thank you for your comments. Our experiment parameters are the same as the simulation parameters in Section 4. In addition, we have also consulted the experiment data and the photos of experiment equipments and added them to the manuscript. Your suggestion are very helpful to us. In future research and writing, we will pay more attention to the recording and collation of experimental details.
Point 2: The simulation in Figure 10 shows that when the DSSS BER is near 10^-4, the MCSK-VTRM is close to the DSSS performance. However, Figure 15 and Table 5 show that the experiment reveals that the MCSK-VTRM performance is closer to the MCSK performance than the DSSS results.
I think that there needs to be some discussion of the discrepancy between simulation and measurement.
Response 2: Thank you for your careful reading. We are very sorry that we have not analysed the reasons for the discrepancy between the experiment results and the simulation results. In the simulation, the BER of the MCSK-VTRM system is almost similar to that of the DSSS system. However, the performance of MCSK-VTRM system cannot be close to that of the DSSS system in the poor experiments. This is because the sparsity of the channel in the experiment is not good enough. As can be seen from Figure 15 in the revised manuscript, the channel in the experiment is multi-path concentrated, and mostly concentrated near the main path. But as shown in Figure 9, the sparsity of channel used in the simulation is good, where most channel coefficients have smaller energy. The distribution of several taps with larger energy is far apart, and the time delay difference between each tap is unequal. From the discussion in Section 3, we can know that the effect of VTRM is related to the multipath structure of the UWA channel. The multipath structure of the channel in the experiment is not ideal enough, so it leads to the results do not reach the ideal state of simulation.

Round 2
Reviewer 2 Report
In this new version, the authors have addressed the points previously raised. I consider that there are enough contributions in this manuscript because of its experimental verification. It deserves to be published and I do recommend the acceptance of this paper. As a "minor correction", I recommend the following:
1)Refs. [4, 5, 6, 7] are missed in the revised version;
2) In Eq. (1), the transmitted signal is expressed as ‘s(t)’, but it also represents the received signal in Eq. (2).
3) Carefully check the paper for spelling and expressions.
Author Response
Thank you very much for your comments concerning our manuscript entitled “M-ary Cyclic Shift Keying Spread Spectrum Underwater Acoustic Communications Based on Virtual Time-Reversal Mirror” (Manuscript ID: sensors-553158).
We went through all the comments carefully and have made corrections exactly according to your kind comments. The main corrections in the manuscript and the response to your comments are as follows:
Point 1: Refs. [4, 5, 6, 7] are missed in the revised version

Response 1: Thank you for your careful reading. We are very sorry that these references are missed. We have added them in Line 43.
Point 2: In Eq. (1), the transmitted signal is expressed as ‘s(t)’, but it also represents the received signal in Eq. (2).
Response 2: Thank you for your careful reading. We have corrected these expressions. In this manuscript, ‘s(t)’ is the transmitted signal. And 'Sr(t)' is the received signal.
Point 3: Carefully check the paper for spelling and expressions.
Response 3: Thank you for your suggestion. We have carefully checked the whole manuscript. And we revised some inappropriate words in Lines 48, 207, 350, 370, 378, and 379.